# Adaptive Shrinkage Estimation for Streaming Graphs

**Nesreen K. Ahmed**
Intel Labs
Santa Clara, CA 95054
nesreen.k.ahmed@intel.com

**Nick Duffield**
Texas A&M University
College Station, TX 77843
duffieldng@tamu.edu

## Abstract

Networks are a natural representation of complex systems across the sciences, and higher-order dependencies are central to the understanding and modeling of these systems. However, in many practical applications such as online social networks, networks are massive, dynamic, and naturally streaming, where pairwise interactions among vertices become available one at a time in some arbitrary order. The massive size and streaming nature of these networks allow only partial observation, since it is infeasible to analyze the entire network. Under such scenarios, it is challenging to study the higher-order structural and connectivity patterns of streaming networks. In this work, we consider the fundamental problem of estimating the higher-order dependencies using adaptive sampling. We propose a novel *adaptive*, *single-pass* sampling framework and unbiased estimators for higher-order network analysis of large streaming networks. Our algorithms exploit adaptive techniques to identify edges that are highly informative for efficiently estimating the higher-order structure of streaming networks from small sample data. We also introduce a novel James-Stein shrinkage estimator to reduce the estimation error. Our approach is fully analytic, computationally efficient, and can be incrementally updated in a streaming setting. Numerical experiments on large networks show that our approach is superior to baseline methods.

## 1   Introduction

Network analysis has been central to the understanding and modeling of large complex systems in various domains, e.g., social, biological, neural, and technological systems [7, 37]. These complex systems are usually represented as a network (graph) where vertices represent the components of the system, and edges represent their direct (observed) interactions over time. The success of network analysis throughout the sciences rests on the ability to describe the complex structure and dynamics of arbitrary systems using only *observed* pairwise interaction data among the components of the system. Many networked systems exhibit rich structural and connectivity patterns that can be captured at the level of pairwise links (edges) or individual vertices. However, higher-order dependencies that capture complex forms of interactions remain largely unknown, since they are beyond the reach of methods that focus primarily on pairwise links. Recently, there has been a surge of studies on higher-order network analysis [4, 9, 52, 43, 20]. These methods focus on generalizing the analysis and modeling of network data from pairwise relationships (e.g., edges) to more complex forms of relationships such as multi-node (many-body) relationships (e.g., motif patterns, hypergraphs) and higher-order network paths that depend on more history [46]. Higher-order connectivity patterns were shown to change node rankings [46, 57], reshape the community structure [52, 9, 56], reveal the hub structure [4], learn more accurate embeddings [42, 41], and generative network models [16].

Many networks are massive, dynamic, and naturally streaming over time [33, 44, 3], with pairwise interactions (i.e., edges that represent communication in the form of user-to-user, user-to-product interactions) are becoming available one at a time in some arbitrary order (e.g., online social networks, Emails, Twitter data, recommendation engines). The massive size and streaming nature of these networks allow only partial observation, since it is infeasible to analyze the entire network. Under

such scenarios, the question of how to study and reveal the higher-order connectivity structure and patterns of streaming networks has remained a challenge. This work is motivated by large-scale streaming network data that are generated by measurement processes (i.e., from online social media, sensors, and communication devices), and we study how to estimate the higher-order connectivity structure of streaming networks under the constraints of partial observation and limited memory. We particularly focus on the estimation of higher-order network patterns captured by small subgraphs, also called network motifs (e.g., triangles or small cliques) [34, 6].

Randomization and sampling techniques are fundamental in the context of graph and matrix approximations in both static and streaming settings; see [33, 29, 26, 5]. The general problem is setup as follows: given a graph $G = (V, K)$ and a budget $m$, find a sampled graph $\widehat{G}$ such that the (expected) number of edges (non-zero entries) is at most $m$ and $\widehat{G}$ is a good proxy for $G$. In the data streaming model, the input graph $G$ is a stream of edges $K = \{k_1 = (u, v), k_2 = (v, w) \dots \}$ and is partially observed as the edges stream and become available to the algorithm one at a time in some arbitrary order. The streaming model is fundamental to applications of online social networks, social media, and recommendation systems where network data become available one at a time (e.g., friendship links, emails, Twitter feeds, user-item preferences, purchase transactions, etc). Moreover, the streaming model is also crucial where network data is streaming from disk storage and random accesses of edges are too expensive. However, the theory and algorithms of current graph sampling techniques are mostly well developed for sampling individual edges to estimate global network properties (e.g., total number of edges in a graph) [25, 50]. Here, we consider instead sampling techniques that can capture how edges connect locally to form small network substructures (i.e., network motifs). Designing new sampling algorithms to estimate the local higher-order connectivity patterns of streaming networks has the potential to improve accuracy and efficiency of sampling and knowledge discovery in streaming networks.

**Contributions.** We propose a novel *topologically adaptive*, *single-pass* priority sampling framework for unbiased estimation of higher-order network connectivity structure of large streaming networks, where edges become available one at a time in some arbitrary order. Specifically, we propose unbiased estimators for *local* counts of subgraphs or motifs containing each edge (Theorem 1) and show how to compute them efficiently for streaming networks (Theorem 2). These estimators are embodied in our proposed adaptive sampling framework (see Algorithm 1).

Our proposed adaptive sampling preferentially selects edges to include in the sample based on their importance weight relative to the variable of interest (i.e., higher-order graph properties), then adapts their weights to allow edges to gain importance during stream processing leading to reduction in estimation variance as compared with static and/or uniform weights.

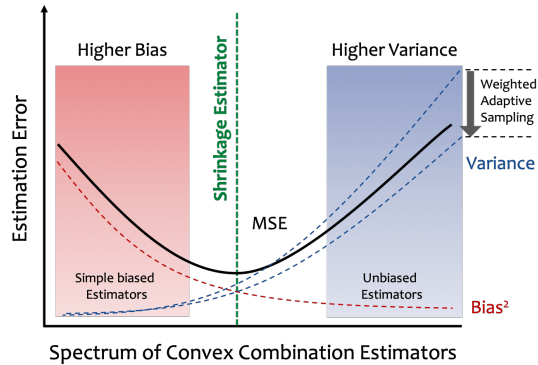

Figure 1: Bias-Variance Trade-off in Graph Sampling

We also propose a novel shrinkage estimator which we formulate as a convex combination estimator to reduce the mean squared error (MSE) (as shown in Figure 1), and we discuss its computation during stream processing (Section 3). Our approach is fully analytic, computationally efficient, and can be incrementally updated as the edges become available one at a time during stream processing. The proposed methods are also generally applicable to a wide variety of networks, including directed, undirected, weighted, and heterogeneous networks.

## 2 Adaptive Sampling Framework

### 2.1 Notation and Problem Definition

Consider an arriving stream $K$ of unique graph edges labelled by the edge identifiers $k \in [|K|]$. Let $G = (V, K)$ denote the undirected graph formed by the edges, where $V$ is the vertex set and $K$ is the edge set. Assume $M$ is a motif (subgraph) pattern of interest, let $H$ denote the class of subgraphs in $G$ that are isomorphic to $M$ (e.g., all triangles or cliques of a given size that appear in $G$). We define the $H$-weighted graph of $G$ as the weighted graph $G_H = (V, K, N)$ with edge

weights $N = \{n_k : k \in K\}$, such that for each edge $k \in K$, $n_k$ is the number of subgraphs in $H$ that are isomorphic to motif $M$ and incident to $k$, i.e., $n_k = |\{h \in H : h \ni k, h \cong M\}|$. We refer to this graph as the *motif-weighted graph*, and we denote A as its motif adjacency matrix [9]. For brevity we will identify a subgraph $h \in H$ with its edge set. Table 3 in the supplementary materials provides a summary of notation. Suppose the edges of $G$ are labelled in some arbitrary order based on their arrival in the stream. Let $G_t = (V_t, K_t)$ denote the subgraph of $G$ formed by the first $t$ edges in this order, $H_t = \{h \in H : h \subset K_t\}$ be the set of subgraphs in $H$ all of whose edges have arrived by $t$, and $(V_t, K_t, N_t)$ be the corresponding $H$-weighted graph of $G_t$ (with weights $N_t = \{n_{k,t} : k \in K_t\}$). This paper studies two questions: (1) how to maintain a reservoir sample $\widehat{K}$ of $m$ edges from the unweighted edge stream $K$, and (2) how to obtain an unbiased estimate of the $H$-weighted graph $G_H = (V_t, K_t, N_t)$ at any time $t \in [|K|]$. We propose a variable-weight adaptive sampling framework for streaming network/graph data, called *adaptive priority sampling*. Our proposed framework preferentially selects edges to include in the sample based on their importance weight, where the weights are relative to the role of these edges in the formation of motifs and general subgraphs of interest (e.g., triangles or small cliques) and can adapt to the changing topology during streaming. Next, we describe the proposed framework (Alg. 1), and discuss its theoretical foundation.

## 2.2 Algorithm Description and Key Intuition

We consider a generic reservoir sample $\widehat{K}$ selected progressively from the edge stream labelled $K = [|K|] = \{1, 2, \ldots, |K|\}$. We assume edges are unique, and therefore they can be identified by their arrival positions (i.e., edge ids); nevertheless we will sometimes emphasize their graph or time aspects, denoting by $k_t$ the edge arriving at time slot $t$, and by $t_k$ the arrival time slot of edge $k$. In Alg. 1, the first $m$ edges are admitted to the sample: $\widehat{K}_t = [t]$ for $t \leq m$. Then, each subsequent edge $t$ is provisionally included in the current sample to form $\widehat{K}'_t = \widehat{K}_{t-1} \cup \{t\}$ (see line 6), from which an edge is *discarded* to produce the sample $\widehat{K}_t$, and maintain the sample size $m = |\widehat{K}_t|$ at any time $t$.

---

**Algorithm 1** Adaptive Priority Sampling (APS)

---

**Input:** Edge stream, sample size $m$, Motif pattern $M$
**Output:** Reservoir Sample $\widehat{K}$

1:   $\widehat{K} \leftarrow \emptyset, z^* \leftarrow 0$          $\triangleright$ Initialize
2:   **for** a new edge $k$ **do**
3:      Generate $u(k) \sim \mathrm{Uni}(0, 1]$
4:      $w(k) \leftarrow \phi$          $\triangleright$ Initial Weight
5:      $p(k) \leftarrow 1$          $\triangleright$ Initial probability
6:      $\widehat{K} \leftarrow \widehat{K} \cup \{k\}$        $\triangleright$ Add $k$ to the sample
7:      // Set of motifs contain $k$ and isomorphic to $M$
8:      $\Delta \leftarrow \{h \subset \widehat{K} : h \ni k, h \cong M\}$
9:      **for** $h \in \Delta$ and $\forall j \in h$ **do**
10:        **if** $z^* > 0$ **then**
11:          $p(j) \leftarrow \min\{p(j), w(j)/z^*\}$, if $j \neq k$
12:        $w(j) \leftarrow w(j) + 1$    $\triangleright$ Update weight for $j$
13:        $p(h) \leftarrow \prod_{j \in h} p(j)$
14:        $n(j) \leftarrow n(j) + 1/p(h)$    $\triangleright$ Update count for $j$
15:        $r(j) \leftarrow w(j)/u(j)$, if $j \neq k$   $\triangleright$ Update Rank for $j$
16:      $r(k) \leftarrow w(k)/u(k)$     $\triangleright$ Rank variable for new edge
17:      **if** $|\widehat{K}| > m$ **then**
18:        $k^* \leftarrow \arg\min_{j \in \widehat{K}} r(j)$
19:        $z^* \leftarrow \max\{z^*, r(k^*)\}$     $\triangleright$ Update threshold
20:        Remove $k^*$ from $\widehat{K}$     $\triangleright$ Discard min rank edge

---

In Algorithm 1, each edge $i \in \widehat{K}'_t$ is assigned a priority *rank* variable defined as $r_{i,t} = w_{i,t}/u_i$, where $w_{i,t}$ is the edge weight at time $t$, and $u_i$ is a uniformly distributed random variable on $(0, 1]$ assigned to the edge on its first arrival. Then, the edge with minimum rank $z_t = \min_{j \in \widehat{K}'_t} r_{j,t}$ is discarded from $\widehat{K}'_t$ to obtain the sample $\widehat{K}_t$ (see lines 17–20). For each edge $i \in \widehat{K}'_t$, we compute the weight $w_{i,t} > 0$ as a function of its previous weight $w_{i,t-1}$ and the sample set $\widehat{K}'_t$.

Upon its arrival, a new edge $k$ is assigned an IID *edge random variable* $u_k$ uniformly distributed on $(0, 1]$, and an initial (constant) weight $\phi$ (lines 3–5), plus the number of target subgraphs/motifs in $\widehat{K}'_t$ that contains $k$ (see lines 9–15). An edge $i \in \widehat{K}'_t$ survives the sampling at time $t$, if and only if there is another edge in $\widehat{K}'_t$ that has the minimum rank, i.e., $r_{i,t} > z_t$.

Conditional on $z_t$, the effective sampling probability of an edge $i \in \widehat{K}_t$ is: $\mathbb{P}\{r_{i,t} > z_t\} = \mathbb{P}\{u_i < w_{i,t}/z_t\} = \min\{1, w_{i,t}/z_t\}$. We note that in the experiments of Section 4, we choose the initial edge weight $\phi = 1$ to be comparable with the edge weight increments due to subgraphs incident to each edge (see line 4). This procedure allows edges to have a chance to be included in the sample with a non-zero probability, regardless of the number of subgraphs incident to them, but not so large as to damp out their topological weight. Next, we discuss how the approach in Algorithm 1 leads to unbiased estimators of general subgraphs/motifs.

### 2.3 Unbiased Estimators of General Subgraphs

Let $S_{i,t}$ denote the arrival of an edge $i$, i.e., $S_{i,t} = I(i \leq t)$. For any subgraph $J \subset K$, where $J$ is a subset of edges (or edge ids), let $S_{J,t} = \prod_{i \in J} S_{i,t}$ indicates whether all edges $i \in J$ have arrived by time $t$, i.e., $S_{J,t} = 1$ if $J \subset K_t$ and 0 otherwise. We observe the local edge count $n_{i,t} = \sum_{J \in H_{i,t}} S_{J,t}$, and $H_{i,t} = \{h \in H_t : h \ni i\}$ is the set of subgraphs (motifs) incident to edge $i$ whose edges have arrived by time $t$.

Theorem 1 establishes unbiased inverse probability estimators [23] for $S_{J,t}$ in the form $\widehat{S}_{J,t} = I(J \subset \widehat{K}_t)/P_{J,t}$ when $t \geq \tau_J := \max_{i \in J} t_i$ (i.e., all edges in $J$ have arrived by time $t$), and $P_{J,t}$ is the sampling probability for the subgraph $J$. For any subgraph $J \subset K$ with $|J| \leq m \leq t$, let $J_t = J \cap [t]$, and define the conditional minimum edge rank over the sample $\widehat{K}'_t$ as $z_{J,t} = \min_{j \in \widehat{K}'_t \setminus J_t} r_{j,t}$. Hence, $z_t = z_{\emptyset,t}$ is the unrestricted minimum rank over $\widehat{K}'_t$. For $i \in J$, we define the edge probabilities $p_{i,t,J}$ to be 1 when $t < i$ and $\min\{1, \min_{i \leq s \leq t} w_{i,s}/z_{J,t}\}$ otherwise. This can be expressed in an iterative form as follows,

$$
p_{i,t,J} = \begin{cases} 1, & \text{if } t < i \\ \min\{p_{i,t-1,J}, w_{i,t}/z_{J,t}\}, & \text{if } t \geq i \end{cases} \tag{1}
$$

We distinguish between $\widetilde{P}_{J,t}$ and $P_{J,t}$. We use $\widetilde{P}_{J,t} = \prod_{i \in J_t} p_{i,t,J}$ to denote the sampling probability of subgraph $J$ at time $t$, conditional on the ranks of edges not in $J$ (i.e., using the conditional min rank $z_{J,t}$). We also use $P_{J,t} = \prod_{i \in J_t} p_{i,t}$, where $p_{i,t} := p_{i,t,\emptyset}$, to denote the sampling probability of subgraph $J$ that employs the threshold $z_t = z_{\emptyset,t}$, i.e., $z_t$ is the unrestricted minimum rank over $\widehat{K}'_t$.

Set $t_J = \min_{i \in J} t_i$, then define $\widetilde{S}_{J_t} = I(J_t \in \widehat{K}_t)/\widetilde{P}_{J,t}$ and the set of variables $\mathcal{Z}_{J,t} = \{z_{J,s} : t_J \leq s \leq t\}$. In Theorem 1, we establish first that $\widetilde{S}_{J,t}$ is an unbiased estimator of $S_{J,t}$, but that estimates can be computed using $\widehat{S}_{J,t}$. This is preferable since $P_{J,t}$ is computed using the unrestricted threshold $z_t$, independent of the subgraph $J$ to be estimated.

**Theorem 1 (Unbiased Subgraph Estimation[1]).**

(I) *The distributions of the edge random variables $\{u_i : i \in J\}$, conditional on $J_t \subset \widehat{K}_t$ and $\mathcal{Z}_{J,t}$, are independent, with each $u_i$ being uniformly distributed on $(0, p_{i,J,t}]$.*

(II) $\mathbb{E}[I(J_t \subset K_t)|\mathcal{Z}_{J,t}, J_{t-1} \subset \widehat{K}_{t-1}] = \widetilde{P}_{J,t}/\widetilde{P}_{J,t-1}$

(III) $\mathbb{E}[\widetilde{S}_{J,t}|\mathcal{Z}_{J,t-1}, J_{t-1} \subset \widehat{K}_{t-1}] = \widetilde{S}_{J,t-1}$, *and hence* $\mathbb{E}[\widetilde{S}_{J,t}] = 1$, *for* $t > t_J$.

(IV) $\widetilde{P}_{J,t} = P_{J,t}$ *when* $J_t \in \widehat{K}_t$ *and hence* $\mathbb{E}[\widehat{S}_{J,t}] = S_{J,t}$, *for all* $t$.

Using Theorem 1, it is straightforward to show that for any edge $i \in \widehat{K}_t$, $\widehat{n}_{i,t} = \sum_{J \in H_{i,t}} \widehat{S}_{J,t}$ is an unbiased estimator of $n_{i,t}$, i.e. $\mathbb{E}[\widehat{n}_{i,t}] = n_{i,t}$.

**Unbiased Estimation from the Last Arriving Edge.** Recall that $\tau_J = \max_{i \in J} t_i$ denotes the time of the last arriving edge $k_{\tau_J}$ of the subgraph $J \subset K$. Set $J^{(0)} = J \setminus \{k_{\tau_J}\}$, and define $\widehat{S}'_{J,t} = \widehat{S}_{J^{(0)},\tau_J - 1}$, where $S'_{J,t}$ indicates subgraph $J$ right before the arrival of the last edge $k_{\tau_J}$.

In Alg. 1, when a new edge arrives at time $t = \tau_J$, Algorithm 1 finds all subgraphs $\Delta \subset H_t$ that are completed by the arriving edge and whose edges are in the sample $\widehat{K}'_t$ (see line 8). For each subgraph $J \in \Delta$ and each edge $i \in J$, we increment the estimate $\widehat{n}_{i,t}$ by the inverse probability $1/P_{J^{(0)},t-1}$, where $P_{J^{(0)},t-1} = \prod_{i \in J^{(0)}} p_{i,t-1}$ is the sampling probability for $S'_{J,t}$ (lines 13–14).

Corollary 1 results from Theorem 1 and establishes that $\mathbb{E}[\widehat{S}'_{J,t}] = 1$, hence, $\widehat{n}_{i,t} = \sum_{J \in H_{i,t}} \widehat{S}'_{J,t}$ is an unbiased estimator for $n_{i,t}$, for all $i \in K_t$. This allows us to update the estimates without risking loss of some edge in $J$ during subsequent sampling (i.e., when the edge with minimum rank is discarded from the sample).

**Corollary 1.** $\mathbb{E}[\widehat{S}'_{J,t}] = 1$ *and hence* $\widehat{n}_{i,t} = \sum_{J \in H_{i,t}} \widehat{S}'_{J,t}$ *is an unbiased estimator of the local subgraph count* $n_{i,t}$ *for all* $i \in K_t$.

## 2.4 Special Case of Non-decreasing Sampling Weights

Computing the probabilities $p_{i,t}$ according to Equation 1 requires an update for each each edge $i \in \widehat{K}_t$ at each time step $t$, i.e., $O(m)$ for each arriving edge. We now show that this computational cost can be reduced when $w_{i,t}$ is non-decreasing in $t$. Let $d_t \leq t$ denote the edge discarded at time $t > m$, i.e., $\{d_t\} = \widehat{K}'_t \setminus \widehat{K}_t$ (line 20 in Alg. 1). We define the sample threshold $z_t^*$ iteratively by $z_m^* = 0$ and $z_t^* = \max\{z_{t-1}^*, z_t\}$, for $t > m$ (see line 19 in Algorithm 1). Define $p_{i,i}^* = \min\{1, w_{i,i}/z_i^*\}$ and $p_{i,t+1}^* = \min\{p_{i,t}^*, w_{i,t+1}/z_{t+1}^*\}$, for $t \geq i$, i.e., similar to Equation 1 but with $z_t$ replaced by $z_t^*$ ( as shown in line 11 in Alg. 1).

**Theorem 2.** *When $w_{i,t}$ is non-decreasing in $t$ then (I) $d_t \neq t$ implies $z_t^* = z_t$; and (II) $p_{i,t}^* = p_{i,t}$ for all $t \geq i$.*

We take advantage of Theorem 2 to reduce the number of updates to the probability $p_{i,t}^*$, Since $w_{i,t}$ is non-decreasing and $z_t^*$ is also non-decreasing, $w_{i,t}/z_t^*$ can only increase when $w_{i,t}$ increases.

During the intervals of constant $w_{i,t}$, $w_{i,t}/z_t^*$ is non-increasing. Therefore, provided that we update $p_{i,t}^*$ at times when $w_{i,t}$ increases, all other updates of $p_{i,t}^*$ can be deferred until needed for estimation (see line 11 of Alg. 1).

**Complexity Analysis.** In Algorithm 1, the sampling reservoir is implemented as a min-heap. Any insertion, deletion, update operation has $O(\log m)$ complexity in the worst case. Retrieving the edge with minimum rank is done in constant time $O(1)$. The complexity of the weight update depends on the target subgraph class, being proportional to the number of edges in new subgraphs created by the arriving edge. In the experiments reported in this paper, the target subgraphs are triangles. For an arriving edge $k = (v_1, v_2)$, the third vertex of any new triangle incident to $k$ lies in the set intersection of the sampled neighbors of $v_1$ and $v_2$ which can be computed in $O(\min\{\deg(v_1), \deg(v_2)\})$, where $\deg(v_1)$ and $\deg(v_2)$ are the sampled vertex degrees of $v_1$ and $v_2$ respectively. This complexity can be achieved if a hash table (or Bloom filter) is used for storing and looping over the sampled neighborhood of the vertex with minimum degree and querying the hash table of the other vertex.

# 3 James-Stein Shrinkage Estimator

It is common in graph sampling to seek unbiased estimators with minimum variance that perform well, e.g., the estimator in Section 2. In this section, we also investigate another desirable estimator, called *shrinkage estimator* [24, 21], that directly reduces the mean squared error (MSE), which is a direct measure of estimation error. In Figure 1, we demonstrate the bias-variance trade-off in graph sampling, which leads to both biased and unbiased estimators. Unbiased estimators of local subgraph counts are subject to high relative variance when the motif counts are small, because in this case the individual count estimates, scaled by the inverse probabilities, are smoothed less by aggregation.

More generally, James and Stein originated the observation that unbiased estimators do not necessarily minimize the mean squared error [24]. In their study, unbiased estimates of high dimensional Gaussian random variables are adjusted through scaling-based regularization and linear combination with dimensional averages. Shrinkage estimation has been used in other settings such as covariance or affinity matrix estimation [45, 55, 11, 28]. Here, we examine shrinkage for the estimated count $\widehat{n}_k$ by convex combination with the *observed* and *un-normalized* count provided by the edge sampling weight $w_k$. By introducing bias through $w_k$, we can obtain further reductions in mean squared error (MSE), additional to the adaptive sampling technique discussed in Section 2.

## 3.1 Optimizing Shrinkage Coefficients

We define a family of shrinkage estimators $\eta = \lambda \widehat{n} + \overline{\lambda} w$, where the *shrinkage coefficient* $\lambda \in [0, 1]$ specifies $\eta$ as a convex combination of the unbiased estimator $\widehat{n} = \widehat{n}_k$ and the un-normalized edge weight $w = w_k$, for any edge $k$. Let $\overline{\lambda}$ denote $1 - \lambda$. The loss $\mathcal{L}(\lambda)$ associated with the shrinkage coefficient $\lambda$ is the mean squared error:

$$\mathcal{L}(\lambda) = \text{Var}(\widehat{\eta}) + (\mathbb{E}[\widehat{\eta}] - n)^2 = \lambda^2 \text{Var}(\widehat{n}) + \overline{\lambda}^2 \text{Var}(w) + 2\lambda\overline{\lambda} \text{Cov}(\widehat{n}, w) + \overline{\lambda}^2 \mathbb{E}[\widehat{n} - w]^2 \quad (2)$$

since $\mathbb{E}[\widehat{\eta} - n] = \mathbb{E}[\widehat{\eta} - \widehat{n}] = \mathbb{E}[\lambda\widehat{n} + \overline{\lambda}w - \widehat{n}] = \overline{\lambda}\mathbb{E}[w - \widehat{n}]$.

$\mathcal{L}$ is convex with derivative $\mathcal{L}'$ specified by,

$$\mathcal{L}'(\lambda)/2 = \lambda \text{Var}(\widehat{n}) - \overline{\lambda} \text{Var}(w) + (1 - 2\lambda) \text{Cov}(\widehat{n}, w) - \overline{\lambda}(\mathbb{E}[\widehat{n} - w])^2 \quad (3)$$

We seek the minimum of $\mathcal{L}$ when $\mathcal{L}'(\lambda) = 0$, i.e., when

$$\lambda = 1 - \frac{\mathrm{Cov}(\widehat{n} - w, \widehat{n})}{\mathbb{E}[(\widehat{n} - w)^2]} = 1 - \frac{\mathrm{Var}(\widehat{n}) - \mathrm{Cov}(\widehat{n}, w)}{\mathbb{E}[(\widehat{n} - w)^2]} \tag{4}$$

We truncate $\overline{\lambda}$ at 1 so that the constraint $\overline{\lambda} \leq 1$ always holds. Since the optimal $\lambda$ is a function of the unknown true covariances, we follow the practice of [12] by employing a plug-in estimator $\widehat{\lambda}$ for $\lambda$ by substituting $(\widehat{n} - w)^2$ in the denominator, and an unbiased estimate for $\mathrm{Cov}(\widehat{n} - w, \widehat{n}) = \mathrm{Var}(\widehat{n}_k) - \mathrm{Cov}(\widehat{n}_k, w_k)$, whose computation we describe next.

## 3.2  Unbiased Estimation of the Variance $\mathrm{Var}(\widehat{n})$

Let $\Delta_{j,t} = H_{j,t} \setminus H_{j,t-1}$ denote the set of subgraphs in $K_t$ that contain an edge $j$ and are completed by the new edge arrival at time $t$. Similarly, let $\widehat{\Delta}_{j,t}$ denote the (possibly empty) set of subgraphs in $\widehat{K}'_t$ that contain an edge $j \in K'_t$ and are completed by the new edge arrival at time $t$. Thus, the estimated count $\widehat{n}_{j,t}$ can be decomposed as: $\widehat{n}_{j,t} = \widehat{n}_{j,t-1} + \sum_{J \in \Delta_{j,t}} \widehat{S}'_{J,t}$.

For any pair of subgraphs $J, L \in H_{j,t}$, the variance of $\widehat{n}_{j,t}$ is specified by:

$$\mathrm{Var}(\widehat{n}_{j,t}) = \sum_{J,L \in H_{j,t}} \mathrm{Cov}(\widehat{S}'_{J,t}, \widehat{S}'_{L,t}) \tag{5}$$

where $\mathrm{Cov}(\widehat{S}'_{J,t}, \widehat{S}'_{L,t})$ is the covariance between two subgraph estimators. Furthermore, the variance $\mathrm{Var}(\widehat{n}_{j,t})$ can also be computed incrementally at each time $t$ as follows,

$$\mathrm{Var}(\widehat{n}_{j,t}) = \mathrm{Var}(\widehat{n}_{j,t-1}) + \sum_{J \in \Delta_{j,t}} \left[ \mathrm{Var}(\widehat{S}'_{J,t}) + 2\,\mathrm{Cov}(\widehat{n}_{j,t-1}, \widehat{S}'_{J,t}) + \sum_{\substack{L \in \Delta_{j,t} \\ L \neq J}} \mathrm{Cov}(\widehat{S}'_{J,t}, \widehat{S}'_{L,t}) \right] \tag{6}$$

where the term $\mathrm{Cov}(\widehat{n}_{j,t-1}, \widehat{S}'_{J,t}) = \sum_{s<t} \sum_{L \in \Delta_{j,s}} \mathrm{Cov}(\widehat{S}'_{J,t}, \widehat{S}'_{L,s})$, for $s < t$.

Theorem 3 is used to establish an unbiased estimator for $\mathrm{Cov}(\widehat{S}'_{J,t}, \widehat{S}'_{L,s})$ in the form,

$$C_{J,t_1;L,t_2} = \widehat{S}'_{J,t_1} \widehat{S}'_{L,t_2} - \widehat{S}'_{J \setminus L, t_1} \widehat{S}'_{L \setminus J, t_2} \widehat{S}'_{J \cap L, t_1 \vee t_2} \tag{7}$$

where $t_1 \geq t_2$, and $t_1 \vee t_2 = \max\{t_1, t_2\}$.

**Theorem 3.** $C_{J,t_1;L,t_2}$ *is an unbiased estimator of* $\mathrm{Cov}(\widehat{S}'_{J,t_1}, \widehat{S}'_{L,t_2})$*, for some time* $t_1 \geq t_2$.

A special case of Theorem 3 happens when $J = L$ and $t_1 = t_2 = t$, which leads to $V(\widehat{S}'_{J,t}) = \widehat{S}'_{J,t}(\widehat{S}'_{J,t} - 1)$, where $V(\widehat{S}'_{J,t})$ is an unbiased estimator of $\mathrm{Var}(\widehat{S}'_{J,t})$.

## 3.3  Unbiased Estimation of the Covariance $\mathrm{Cov}(\widehat{n}, w)$

Following the notation in Section 3.2, for each edge $j$, the weight $w_{j,t}$ is a random quantity incremented by 1 for each subgraph $J \in \Delta_{j,t}$ completed by the new edge arrival at time $t$. Thus, $w_{j,t}$ can be written as a sum of random counts, i.e., un-normalized indicator functions analogous to how $\widehat{n}_{j,t}$ is written as a sum of inverse probability estimators. Let $I_{J,t} = I(J \subset \widehat{K}_t)$ be the indicator of subgraph $J$, and recall that $J^{(0)}$ is the subgraph $J$ without the last arriving edge $k_{\tau_J}$. Define $I'_{J,t} = I_{J_0, \tau_J - 1}$, i.e., the indicator that all edges but the final edge are present in the sample $\widehat{K}_{t-1}$ immediately before the arrival of the final edge ($k_{\tau_J}$ of $J$). When the new edge $k_{\tau_J}$ arrives at time $t = \tau_J$, each edge in $J^{(0)}$ has its weight incremented; see line 12 of Algorithm 1. Thus, we can write $w_{j,t} = \sum_{J \in H_{j,t}} I'_{J,t}$, analogous to Corollary 1, and decompose $w_{j,t} = w_{j,t-1} + \sum_{J \in \Delta_{j,t}} I'_{J,t}$.

Computing the optimal skrinkage $\lambda$ estimator in Equation 4 requires estimates of the covariance $\mathrm{Cov}(\widehat{n}_{j,t}, w_{j,t})$ for each edge $j \in \widehat{K}_t$, which is estimated in turn and follow by linearity from the estimates of the covariance $\mathrm{Cov}(\widehat{S}'_{J,t}, I'_{J,t})$. Theorem 4 establishes an unbiased estimator for the

general case of $\mathrm{Cov}(\widehat{S}'_{J_1,t_1}, I'_{J_2,t_2})$, when $t_1 \geq t_2$. Lemma 1 is central to both the proof of Theorem 4 and the computation of covariance estimates[2].

**Lemma 1.** *For* $J_1 \cap J_2 = \emptyset$ *and* $t_1 \geq t_2$, *then* $\mathbb{E}[\widehat{S}'_{J_1,t_1} I'_{J_2,t_2}] = \mathbb{E}[I'_{J_2,t_2}]$ *and hence* $\mathrm{Cov}(\widehat{S}'_{J_1,t_1}, I'_{J_2,t_2}) = 0$.

**Theorem 4** (**Unbiased Subgraph Covariance Estimation**)**.**

(I) *When* $t_1 \geq t_2$, $\mathrm{Cov}(\widehat{S}'_{J_1,t_1}, I'_{J_2,t_2})$ *has unbiased estimator* $D_{J_1,t_1;J_2,t_2} = \widehat{S}'_{J_1,t_1} I'_{J_2,t_2} - \widehat{S}'_{J_1 \setminus J_2,t_1} \widehat{S}'_{J_1 \cap J_2,t_1 \vee t_2} P_{J_1 \cap J_2,t_2} I'_{J_2 \setminus J_1,t_2}$.

(II) $D_{J_1,t_1;J_2,t_2} > 0$ *iff* $\widehat{S}'_{J_1,t_1} > 0$ *and* $I'_{J_2,t_2} > 0$. *Hence* $D_{J_1,t_1;J_2,t_2}$ *can be computed from samples that have been taken.*

(III) *For the special case* $J_1 = J_2 = J$ *and* $t_1 = t_2 = t$ *then* $D_{J,t;J,t} = \widehat{S}'_{J,t} \overline{P}_{J,t} = I'_{J,t}(P_{J,t}^{-1} - 1)$.

# 4 Experiments & Discussion

**Experimental Setup.** We test on graphs from different domains and with different characteristics; see [40] for data downloads. Table 1 provides a summary of dataset characteristics, where $|V|$ is the number of vertcies, $|K|$ is the number of edges, $T$ is the number of triangles, and $T_{\max}$ is the maximum triangle count per edge. For all graph datasets, we consider an undirected, unweighted, simplified graph without self loops.

Table 1: Summary of Graph Statistics

| graph | $|V|$ | $|K|$ | $T$ | $T_{\max}$ |
|---|---|---|---|---|
| SOC-FLICKR | 514K | 3.2M | 58.8M | 2236 |
| SOC-LIVEJOURNAL | 4.03M | 27.9M | 83.6M | 586 |
| SOC-YOUTUBE | 1.13M | 2.98M | 3.05M | 4034 |
| WIKI-TALK | 2.4M | 4.7M | 9.2M | 1631 |
| WEB-BERKSTAN-DIR | 685K | 6.7M | 64.7M | 45057 |
| CIT-PATENTS | 3.8M | 16.5M | 7.5M | 591 |
| SOC-ORKUT-DIR | 3.07M | 117.2M | 627.6M | 9145 |

Edge streams are obtained by randomly permuting the edges in each graph, and the same edge order is used for all the methods. We repeat the experiment ten different times with sample fractions $f = \{0.10, 0.20, 0.40, 0.50\}$. All experiments were performed using a server with two Intel Xeon E5-2687W 3.1GHz CPUs, 256GB of memory. The experiments are executed independently for each sample fraction. Additional results and ablation studies are discussed in the supplementary materials. Our experimental setup is summarized as follows:

- For each sample fraction, we use Algorithm 1 to collect a sample $\widehat{K}$, from edge stream $K$.
- The experiments in this section use triangles as an example of the motif pattern $M$. However, the approach itself is general and applicable to any motif patterns.
- During stream processing, we compute unbiased estimators and James-Stein shrinkage estimators of the local triangle counts for the sampled edges, as discussed in Sections 2–3.
- Given a sample $\widehat{K} \subset K$, we compute the mean squared error (MSE), and the relative spectral norm [1], $\|A - \widehat{A}\|_2/\|A\|_2$, where A is the exact triangle-weighted adjacency matrix of the input graph, $\widehat{A}$ is the average estimated triangle-weighted adjacency matrix of the sampled graph, and $\|A\|_2$ is the spectral norm of A.
- We compare the results of Algorithm 1 with uniform sampling (i.e., reservoir sampling [53]) using the Horvitz-Thompson estimator, and we also compare with Triest sampling [48]. All baseline methods use the same experimental setup as the proposed method.

## 4.1 Comparison to Baseline Methods

We collect a sample of edges $\widehat{K} \subset K$ from the edge stream $K$ in a single pass, which we use to construct the motif-weighted graph, where $M$ is the triangle motif and A is adjacency matrix of the triangle-weighted graph. We use $\widehat{A}$ to denote the estimator of A obtained by sampling. We compute the shrinkage estimator as discussed in Section 3. And, we report the MSE at sample fraction $f = 0.20$ in Table 2, which demonstrates the following insight: the shrinkage estimator applied to adaptive priority (APS) sampling significantly improves the performance of the vanilla APS which

Table 2: MSE and Relative Spectral Norm at sampling fraction $f = 0.2$. APS: Adaptive Sampling, APS JS: APS with shrinkage Estimation, Unif: Uniform Sampling, Triest: Triest Sampling.

| | Mean Squared Error (MSE) | | | | $\|A - \widehat{A}\|_2/\|A\|_2$ | | | |
|---|---|---|---|---|---|---|---|---|
| **graph** | APS | APS JS | Unif | Triest | APS | APS JS | Unif | Triest |
| SOC-FLICKR | 22.30K | **295.13** | 6.3K | 7.46K | 0.5793 | **0.0478** | 0.4321 | 0.5149 |
| SOC-LIVEJOURNAL | 214.80 | **16.11** | 257.60 | 293.67 | 0.0269 | **0.0089** | 0.429 | 0.5092 |
| SOC-YOUTUBE-SNAP | 11.35 | **6.68** | 119.79 | 145.87 | **0.0455** | 0.079 | 0.4159 | 0.4982 |
| WIKI-TALK | 7.70 | **5.32** | 589.92 | 680.67 | **0.0105** | 0.0359 | 0.4315 | 0.5109 |
| WEB-BERKSTAN-DIR | 7.32K | **561.20** | 10.70K | 14.03K | 0.1169 | **0.0557** | 0.4381 | 0.6163 |
| CIT-PATENTS | 6.02 | **3.03** | 10.59 | 10.91 | **0.0187** | 0.0428 | 0.4325 | 0.4914 |
| SOC-ORKUT-DIR | 2.08K | **70.79** | 467.90 | 613.89 | 0.1086 | **0.0726** | 0.4385 | 0.4241 |

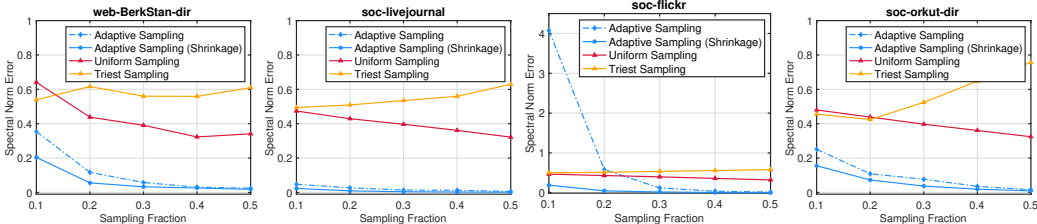

Figure 2: Relative spectral norm $\|A - \widehat{A}\|_2/\|A\|_2$ versus the sampling fraction using all sampling methods. Notably, APS and APS with shrinkage converge faster than uniform and Triest sampling

uses Horvitz-Thompson estimator for all graphs. This is particularly clear for soc-flickr and soc-orkut for which the APS shrinkage (APS JS) significantly outperforms all the other methods.

We also consider the spectral norm as another measure of approximation quality in addition to MSE. The spectral norm $\|A - \widehat{A}\|_2$ was previously used for matrix approximation [1]. $\|A - \widehat{A}\|_2$ measures the strongest linear trend of $A$ that is not captured by the estimator $\widehat{A}$. This is different from the mean squared error which focused on the magnitude of the estimates.

We report the relative spectral norm (i.e., $\|A - \widehat{A}\|_2/\|A\|_2$) at sample fraction $f = 0.20$ for various graphs in Table 2. The experiments demonstrate that for all of the example graphs, both APS and APS with shrinkage significantly outperform uniform reservoir sampling and Triest sampling. One observed exception is the soc-flickr graph, where the estimates using APS is significantly high due to the high variance of Horvitz-Thompson estimation for edges with small counts. Under such scenarios, the APS with shrinkage significantly helps and improves the original APS estimates. We also notice the difference between how the MSE ranks the best methods versus the relative spectral norm. A good example of this is the soc-orkut graph, for which APS performs worse than the baselines. However, APS is superior to uniform sampling and Triest sampling for the relative spectral norm. Thus, despite of the large mean squared error, APS (even without shrinkage) captures the linear trend and structure of the data better than uniform reservoir sampling and Triest sampling. Finally, Figure 2 shows the convergence performance of relative spectral norm as a function of the sampling fraction. Notably, APS and APS with shrinkage converge faster than uniform and Triest sampling, and we observe that shrinkage estimation significantly improves the vanilla APS.

## 4.2 Analysis of the Estimated Distribution

We take the top-k non-zero *edge weights* of the exact triangle-weighted adjacency matrix A, and we compare them against their corresponding estimates obtained by sampling. Figures 3 shows the top-1M weights for APS with shrinkage estimation. Similar figures for uniform sampling and Triest sampling are reported in Section D of the supplementary materials (Fig 8 and Fig 9 respectively). The results demonstrate the more accurate performance of APS with shrinkage estimation compared to the baseline methods; more specifically, APS with shrinkage estimation preserves the distribution and ranks of the top-k edge weights compared to uniform and Triest sampling. We report the analysis for two sampling fractions $f = \{0.20, 0.40\}$.

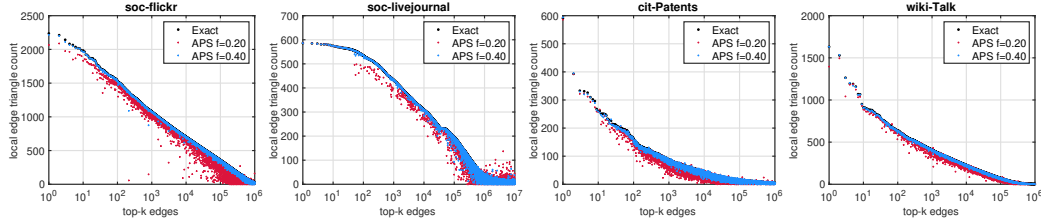

Figure 3: Each Plot corresponds to one graph at sampling fractions $f = \{0.20, 0.40\}$, and shows the estimated weight of the top-1M edges using APS with Shrinkage Estimation vs the exact edge weight. The top-1M edges are ranked based on their true triangle counts.

In Figure 4, we compare APS against APS with shrinkage estimation for the soc-livejournal graph. The results show how the shrinkage estimator reduces the variance of APS, in particular for small local counts with high variance (i.e., as observed in the tail of the edge weight distribution). In Section C in the supplementary materials, we discuss an ablation study of Algorithm 1.

# 5   Related Work

Here, we categorize the related work in three research areas: (1) Higher-order Network Analysis, (2) Graph Approximation, and (3) IID Stream Sampling.

*Higher-order Network Analysis.* There has been an increasing interest in higher-order network analysis and modeling in particular to generalize pairwise links to many-body relationships with arbitrary node sets and motifs; see [34, 9, 52, 56, 4, 54, 43, 20, 41, 16]. The majority of these methods focus on small static networks that fit in memory.

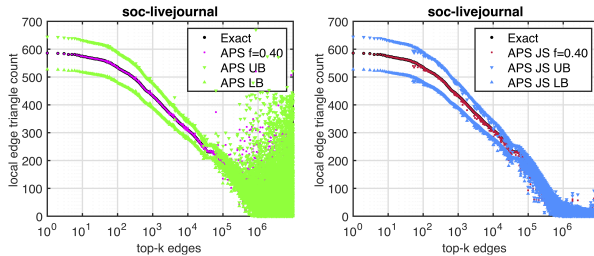

Figure 4: Distribution of the soc-livejournal graph using sampling fraction $f = 0.4$. Left: APS estimated vs exact distribution. Right: APS with Shrinkage estimator (James-Stein JS) vs exact distribution. UB: upper bound, LB: lower bound.

*Graph Approximation.* Randomization in the context of graph approximation is a well-studied topic; see [13, 22, 29, 49] and [33, 3] for a survey. Much work was devoted for triangle count approximation and other motifs for static graphs (see [10, 51, 47, 50, 17, 38]) and for streaming graphs (see [8, 48, 5, 25, 32, 2]). In the streaming setting, most work focused on estimating point statistics using fixed probabilities, e.g., the global triangle or motif count using reservoir based sampling approaches; see [53]. In this paper, we focus instead on estimating the motif-weighted graph from a stream of unweighted edges, and propose a general novel methodology for adaptive priority sampling with shrinkage estimation. We compare against the state-of-the-art approach, Triest sampling [48] and we obtain significant improvement over their method. Triest sampling maintains a sample of edges from the stream using reservoir sampling [53] and random pairing [18] to exploit the available memory as much as possible. However, our approach provides a sampling framework in which edges are included in the reservoir sample based on their importance and topological relevance in the formation of local motifs and subgraphs of interest, and edge weights are allowed to adapt to the changing topology of the reservoir sample.

*IID Stream Sampling.* Prior work focused on IID streams (e.g., IP networks, DB transactions, etc), e.g., single-pass reservoir sampling ([27, 36, 53]), order and threshold sampling ([14, 39, 15]), and probability proportional to size sampling (IPPS). These methods were designed for sampling IID data streams (e.g., IP networks, DB transactions, etc). Here, we focus instead on streaming graphs (non-iid data). Thus, the prior work on IID streams cannot be directly applied in this setting where the focus is on higher-order subgraphs, and extending these methods to non-IID streams is subject to further research.

## Broader Impact

There is a burgeoning recent literature of statistical estimation and adaptive data analysis of the higher-order structural properties of graphs in both the streaming and non streaming context that reflect the importance and interest of this topic for the graph algorithms and relational learning research community. On the other hand, shrinkage estimators are an established technique from more general statistics. This paper is the first to apply shrinkage based methods in the context of graph approximation. The expected broader impact is as a proof of concept that shows the way for other researchers in this area to improve estimation quality. Moreover, this work fits under statistical inference for temporal relational/network data, which would enable statistical analysis and learning for network data that appear in streaming settings, in particular when exact solutions are not feasible (similar to the important literature on randomization algorithms for data matrices [1]).

Furthermore, there are many applications where the data has a pronounced temporal, relational, and spatial structure (e.g., relational data). Examples of Non-IID streams include (i) non-independence due to temporal clustering in communication graphs on internet, online social networks, physical contact networks, and social media such as flash crowds and coordinated botnet activity; (ii) non-identical distributions in activity on these networks due to diurnal and other seasonal variations, synchronization of user network activity e.g., searches stimulated by hourly news reports. The proposed framework is suitable for these applications, because it makes no statistical assumptions concerning the arrival stream and the order of the arriving edges.

## Acknowledgments

Nick Duffield is supported by the National Science Foundation under awards ENG-1839816, IIS-1848596 and CCF-1934904.

## Footnotes

[1]Proofs of all the theorems are discussed in the supplementary materials.

[2]The computational details and proofs for shrinkage estimation are discussed with examples in the supplementary materials

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
