[Supplementary Material]

# A  Theorem Proofs

Table 3: Summary of Notation

| Notation | Description |
|---|---|
| $k_t$ (or just $t$) | Edge arriving at time $t$ |
| $\widehat{K}_t$ | Sample set after edge $t$ processed |
| $\widehat{K}_t'$ | Edges in reservoir prior to selection at time $t$ |
| $J$ | Generic edge subset |
| $J_t$ | Edges from $J$ that have arrived by $t$ |
| $S_{J,t}$ | Indicator variable that indicates if all edges in $J$ have arrived by $t$ |
| $\widehat{S}_{J,t}$ ($\widehat{S}'_{J,t}$) | Inverse probability estimator of $S_{J,t}$ (estimator without last arriving edge) |
| $I_{J,t}$ ($I'_{J,t}$) | Un-normalized estimator of $S_{J,t}$ (estimator not using last arriving edge) |
| $w_{i,t}$ | Weight of edge $i$ at time $t \geq i$ |
| $u_i$ | IID uniform $(0,1]$ variable for edge $i$ |
| $r_{i,t}$ | Priority rank variable of edge $i$ at time $t \geq i$ |
| $z_{J,t}$ | Minimum priority rank of non-$J$ edges prior to $t$ |
| $z_t$ | $z_{\emptyset,t}$ i.e., unrestricted minimum priority rank |
| $z_t^*$ | Cumulative maximum of $z_{t'}$ for $t' \leq t$ |
| $H$ ($H_t$) ($H_{k,t}$) | Set of motifs (those with all their edges arrived by $t$) (also containing edge $k$) |
| $n_{k,t}$ | Total number of members of $H_t$ than contain $k$ |
| $\widehat{n}_{k,t}$ | Estimator of $n_{k,t}$ |
| $\eta$ | Generic James-Stein estimator for an edge count $n$ |
| $\lambda$ | Mixture parameter (i.e., shrinkage coefficient) in $\eta$ |
| $p_{i,t}$ | Probability of inclusion of edge $i \in \widehat{K}_t$ at $t \geq i$ |
| $P_{J,t}$ | Probability of inclusion of edges from $J_t$ in $\widehat{K}_t$ at time $t \geq i$ |
| $t_J$ | Minimum time over all edges in $J$, i.e. $\min_{i \in J} t_i$ |
| $\tau_J$ | Time of the last arriving edge in $J$, $\max_{i \in J} t_i$ |

**Proof of Theorem 1.**

*Proof.* Any subgraph $J$ can be defined as a subset of edges from the set of all edges $K$. Suppose $J_t \subset \widehat{K}_t'$, then $J_t$ survives the sampling at time $t$ (i.e., $J_t \subset \widehat{K}_t$), if and only if another edge $j \in \widehat{K}_t' \backslash J_t$ has minimum rank $z_{J,t} = \min_{j \in \widehat{K}_t' \backslash J_t} r_{j,t}$, i.e., if $r_{i,t} > z_{J,t}$, or equivalently, $u_i < w_{i,t}/z_{J,t}$ for all $i \in J_t$. Denote $A_{i,J,s} = \{u_i < w_{i,s}/z_{J,s}\}$ as the event when $i \in J_s \cap \widehat{K}_s$. Then for $t_J \leq \tau_J \leq t$, the event $\{J \subset \widehat{K}_t\}$ decomposes as $\bigcap_{t_J \leq s \leq t} B_{J,s}$ where $B_{J,s} = \bigcap_{i \in J_s} A_{i,J,s}$.

(I) The proof is by induction on $t$. For $t < t_J$ the conditioning is trivial and $u_i$ are IID on $(0,1] = (0, p_{i,J,t}]$. The same property holds at general $t$ for all $i \in J$ which have not yet arrived, i.e., for $i \in J \backslash J_t$. Consider now $t \geq t_J$ and assume that the result holds for $t-1$. The weights $w_{i,t}$ for $i \in J_t \cap \widehat{K}_t'$ are fixed by the conditioning on the event $\{J_{t-1} \subset \widehat{K}_{t-1}\}$. Further conditioning on $z_{J,t}$ and $J_t \subset \widehat{K}_t$ requires $u_i < w_{i,t}/z_{J,t}$ for all $i \in J_t \subset \widehat{K}_t$. Imposing this condition on the assumed independent uniform distributions of $u_i$ on $(0, p_{i,J,t-1}]$ results in independent uniform distributions of $u_i$ on $(0, \min\{p_{i,J,t-1}, w_{i,t}/z_{J,t}\}] = (0, p_{i,J,t}]$.

(II) The conditional expectation of the indicator $I(J_t \subset \widehat{K}_t)$ is,

$$\mathbb{E}[I(J_t \subset \widehat{K}_t)|\mathcal{Z}_{J,t}, \ J_{t-1} \subset \widehat{K}_{t-1}]$$
$$= \ \mathbb{P}[B_{J,t}|\mathcal{Z}_{J,t}, \ J_{t-1} \subset \widehat{K}_{t-1}]$$
$$= \ \mathbb{P}[\cap_{i \in J_t}\{u_i < w_{i,t}/z_{J,t}\}|\mathcal{Z}_{J,t}, \ J_{t-1} \subset \widehat{K}_{t-1}]$$
$$= \ \widetilde{P}_{J,t}/\widetilde{P}_{J,t-1} \qquad (8)$$

where in the last step we have used the statement of part (I) for the distribution of $u_i$ conditioning on $\mathcal{Z}_{J,t}$ and $\{J_{t-1} \subset \widehat{K}_{t-1}\}$, since $w_{i,t}$ is assumed determined given $\widehat{K}_{t-1}$.

(III) By using (II), we find that the conditional expectation of $\widetilde{S}_{J,t}$ is:

$$\mathbb{E}[\widetilde{S}_{J,t}|\mathcal{Z}_{J,t}, J_{t-1} \subset \widehat{K}_{t-1}] = \frac{1}{\widetilde{P}_{J,t}}\mathbb{E}[I(J_t \subset \widehat{K}_t)|\mathcal{Z}_{J,t}, J_{t-1} \subset \widehat{K}_{J,t-1}]$$

$$= \widetilde{S}_{J,t-1} \tag{9}$$

which is independent of the conditioning on $z_{J,t}$ and hence,

$$\mathbb{E}[\widetilde{S}_{J,t}|\mathcal{Z}_{J,t-1}, J_{t-1} \subset \widehat{K}_{t-1}] = \widetilde{S}_{J,t-1} \tag{10}$$

The initial value (for the first edge arrival at time $t_J$) is $\widetilde{S}_{J,t_J} = I(t_J \in \widehat{K}_{t_J})/p_{t_J,J,t_J} = I(u_{t_J} < w_{t_J,t_J}/z_{J,t_J})/p_{t_J,J,t_J}$. Clearly $\mathbb{E}[\widetilde{S}_{J,t_J}|z_{J,t_J}] = 1$ and hence $\mathbb{E}[\widetilde{S}_{J,t_J}] = 1$. Finally $\mathbb{E}[\widetilde{S}_{J,t}] = 1$ for all $t \geq t_J$ by chaining the conditional expectations.

(IV) Trivially $\widehat{S}_{J,t} = S_{J,t} = 0$ for $t < \tau_J$. Since $z_{J,t} = z_t$ when $J \subset \widehat{K}_t$, $P_{J,t} = \widetilde{P}_{J,t}$ and hence $\widehat{S}_{J,t} = \widetilde{S}_{J,t}$ for $t \geq \tau_J$ and $\mathbb{E}[\widehat{S}_{J,t}] = 1$ by (III).

□

**Proof of Theorem 2.**

*Proof.* (I) If $d_t \neq t$, $t$ is admitted to the sample and hence

$$z_t = \frac{w_{d_t,t}}{u_{d_t}} \geq \frac{w_{d_t,s}}{u_{d_t}} > z_s \tag{11}$$

for all $s \in [d_t, t]$. Since edge $d_t$ is discarded at time $t$, and $d_t \neq t$, then the minimum rank $z_t = r_{d_t,t} = w_{d_t,t}/u_{d_t}$.

The first inequality follows from the non-decreasing property of $w_{d_t,t}$. The second inequality follows since edge $d_t$ survives the sampling from time $d_t$ until $t$ and hence its rank cannot be lower than the threshold $z_s$ for any $s$ in that interval. But since the edge $d_t$ was admitted to the sample at time, we have $d_{d_t} \neq d_t$, where $d_{d_t}$ is the discarded edge at time $d_t$. Hence, we apply the argument back recursively to the first sampling time. Hence, $z_t^* = \max\{z_{t-1}^*, z_t\} = z_t$.

(II) By assumption if an edge $i$ is admitted to $\widehat{K}_i$, then $i \neq d_i$ and so by (I) and Equation 1, $p_{i,i} = \min\{1, w_{i,i}/z_i\} = \min\{1, w_{i,i}/z_i^*\} = p_{i,i}^*$. The general case is by induction. Assume $p_{i,s} = p_{i,s}^*$ for all times $s \in [i, t]$, and $z_{t+1} > z_t^*$, then $z_{t+1}^* = z_{t+1}$ hence $p_{i,t+1}^* = p_{i,t+1}$. If $z_{t+1} \leq z_t^*$, then $z_{t+1}^* = z_t^*$ and hence

$$\frac{w_{i,t+1}}{z_{t+1}} \geq \frac{w_{i,t+1}}{z_{t+1}^*} \geq \frac{w_{i,t}}{z_{t+1}^*} = \frac{w_{i,t}}{z_t^*} \tag{12}$$

Thus we can replace $z_{t+1}$ by $z_{t+1}^*$ in (1) but use of either leaves the iterated value unchanged, since by the induction hypothesis, both are greater than $p_{i,t} \leq w_{i,t}/z_t^*$

□

**Proof of Theorem 3.**

*Proof.*

$$\text{Cov}(\widehat{S}'_{J,t_1}, \widehat{S}'_{L,t_2}) = \mathbb{E}[\widehat{S}'_{J,t_1}\widehat{S}'_{L,t_2}] - \mathbb{E}[\widehat{S}'_{J,t_1}]\mathbb{E}[\widehat{S}'_{L,t_2}]$$

$$= \mathbb{E}[\widehat{S}'_{J,t_1}\widehat{S}'_{L,t_2}] - 1 \tag{13}$$

From Theorem 1, and since $J \setminus L$, $L \setminus J$, and $J \cap L$ are disjoint subsets, we have,

$$\mathbb{E}[\widehat{S}'_{J \setminus L,t_1}\widehat{S}'_{L \setminus J,t_2}\widehat{S}'_{J \cap L,t_1 \vee t_2}] = 1 \tag{14}$$

Thus, $\mathbb{E}[C_{J,t_1;L,t_2}] = \text{Cov}(\widehat{S}'_{J,t_1}, \widehat{S}'_{L,t_2}) = \mathbb{E}[\widehat{S}'_{J,t_1}\widehat{S}'_{L,t_2}] - 1$.

A special case of Theorem 3 happens when $J = L$ and $t_1 = t_2 = t$, which leads to $V(\widehat{S}'_{J,t}) = \widehat{S}'_{J,t}(\widehat{S}'_{J,t} - 1)$, where $V(\widehat{S}'_{J,t})$ is an unbiased estimator of $\text{Var}(\widehat{S}'_{J,t})$.

□

(a) Two Disjoint Subgraphs (triangles)

(b) Two Overlapping Subgraphs (triangles)

Figure 5: Illustrative Example of Disjoint and Overlapping Triangles

**Proof of Lemma 1.**

*Proof.* Let $J = J_1 \cup J_2$. Chaining conditional expectations from Theorem 1(III)

$$\mathbb{E}[\widehat{S}'_{J_1,t_1} I'_{J_2,t_2} | \mathcal{Z}_{J,t_2}, J_{t_2-1} \subset \widehat{K}_{t_2-1}]$$
$$= \mathbb{E}[\widehat{S}'_{J_1,t_2} I'_{J_2,t_2} | \mathcal{Z}_{J,t_2}, J_{t_2-1} \subset \widehat{K}_{t_2-1}]$$
$$= \frac{1}{P_{J_1,t_2}} \mathbb{P}[\cap_{i \in J}\{u_i < w_{i,t_2}/z_{J,t_2}\} | \mathcal{Z}_{J,t_2}, J_{t_2-1} \subset \widehat{K}_{t_2-1}]$$
$$= \mathbb{P}[\cap_{i \in J_2}\{u_i < w_{i,t_2}/z_{J,t_2}\} | \mathcal{Z}_{J,t_2}, J_{t_2-1} \subset \widehat{K}_{t_2-1}]$$
$$= \mathbb{E}[I'_{J_2,t_2} | \mathcal{Z}_{J,t_2}, J_{t_2-1} \subset \widehat{K}_{t_2-1}] \tag{15}$$

using Theorem 1(I). Hence $\mathbb{E}[\widehat{S}'_{J_1,t_1} I'_{J_2,t_2}] = \mathbb{E}[I'_{J_2,t_2}]$ and since $\mathbb{E}[\widehat{S}'_{J_1,t_1}] = 1$, then the $\text{Cov}(\widehat{S}'_{J_1,t_1}, I'_{J_2,t_2}) = \mathbb{E}[\widehat{S}'_{J_1,t_1} I'_{J_2,t_2}] - \mathbb{E}[\widehat{S}'_{J_1,t_1}] \mathbb{E}[I'_{J_2,t_2}] = 0$. □

**Proof of Theorem 4.**

*Proof.* (I) Since $\mathbb{E}[\widehat{S}'_{J_1,t_1}] = 1$ it suffices to show that (the negative of) the second term in the definition of $D_{J_1,t_1;J_2,t_2}$ in Theorem 4(i) has expectation $\mathbb{E}[I'_{J_2,t_2}]$. When $t_1 \geq t_2$ then repeating the conditioning argument of Lemma 1, this term has conditional expectation

$$\mathbb{E}[\widehat{S}'_{J_1,t_1} P_{J_1 \cap J_2,t_2} I'_{J_2 \setminus J_1,t_2} | \mathcal{Z}_{J,t_2}, J_{t_2-1} \subset \widehat{K}_{t_2-1}]$$
$$= \mathbb{E}[\widehat{S}'_{J_1,t_2} P_{J_1 \cap J_2,t_2} I'_{J_2 \setminus J_1,t_2} | \mathcal{Z}_{J,t_2}, J_{t_2-1} \subset \widehat{K}_{t_2-1}]$$
$$= \mathbb{E}[I'_{J_2,t_2} | \mathcal{Z}_{J,t_2}, J_{t_2-1} \subset \widehat{K}_{t_2-1}] \tag{16}$$

and hence the stated property holds.

(II) Holds since $\widehat{S}'_{J,t} > 0$ implies $I'_{J,t'} > 0$ for $t \geq t' \geq t_J$

(III) is a special case of (I).

□

# B  Example: Estimators for Local Triangle Counts

Assume the motif $M$ is a triangle in the form $J = (i,j,k)$, where the edges in the triangle are ordered by their arrival times, i.e., $i < j < k$. Let $k$ denote the new edge arriving at time $t$, and $\widehat{\Delta}_t = \{J = (i,j,k) \subset \widehat{K}'_t\}$ be the set of new triangles completed by $k$ at time $t$. We now show how the estimators can be incremented for each triangle. Note that edges $i,j \in \widehat{K}'_t$ can participate in only one triangle at time $t$.

**Unbiased estimator for $\widehat{n}$.** By applying Theorem 1, each triangle $J = (i,j,k) \in \widehat{\Delta}_t$ results in an increment of $\widehat{S}'_{J,t} = 1/(p_{i,t}p_{j,t})$ in the count estimator for each edge in the triangle as follows:

$$\widehat{n}_i \leftarrow \widehat{n}_i + 1/(p_{i,t}p_{j,t})$$
$$\widehat{n}_j \leftarrow \widehat{n}_j + 1/(p_{i,t}p_{j,t})$$
$$\widehat{n}_k \leftarrow \widehat{n}_k + 1/(p_{i,t}p_{j,t})$$

**Unbiased estimator for $\mathrm{Var}(\widehat{n})$.** By applying Theorem 3, each triangle $J = (i, j, k) \in \widehat{\Delta}_t$ results in an increment of $\mathrm{Var}(S'_{J,t}) = \big(1/(p_{i,t}p_{j,t}) - 1\big)/(p_{i,t}p_{j,t})$ in the variance estimator of the count for each edge in the triangle as follows:

$$\mathrm{Var}(\widehat{n}_i) \leftarrow \mathrm{Var}(\widehat{n}_i) + \big(1/(p_{i,t}p_{j,t}) - 1\big)/(p_{i,t}p_{j,t})$$
$$\mathrm{Var}(\widehat{n}_j) \leftarrow \mathrm{Var}(\widehat{n}_j) + \big(1/(p_{i,t}p_{j,t}) - 1\big)/(p_{i,t}p_{j,t})$$
$$\mathrm{Var}(\widehat{n}_k) \leftarrow \mathrm{Var}(\widehat{n}_k) + \big(1/(p_{i,t}p_{j,t}) - 1\big)/(p_{i,t}p_{j,t})$$

**$\mathbf{Cov}(\widehat{S}'_{J,t}, \widehat{S}'_{L,s})$.** By applying Theorem 3, we detail all the possible cases for the computation of the covariance $\mathrm{Cov}(\widehat{S}'_{J,t}, \widehat{S}'_{L,s})$, where $L = (i', j', k')$ is another triangle, and $L \neq J$:

1. $J \cap L = \emptyset$: if the two triangles are disjoint, then $\mathrm{Cov}(\widehat{S}'_{J,t}, \widehat{S}'_{L,s}) = 0$, see Figure 5 for an example.

2. $s = t$: assume $L = (i', j', k) \in \widehat{\Delta}_t$ is another triangle completed by $k$, and $L \neq J$. This means that $J \cap L = \{k\}$, (see Figure 5), and $\widehat{S}'_{J\cap L, t\vee S} = 1$. Then, the estimator of the covariance $\mathrm{Cov}(\widehat{S}'_{J,t}, \widehat{S}'_{L,s}) = 0$.

3. $s < t$: assume $L = (i', j', k_s) \in \widehat{\Delta}_s$ is another triangle completed by edge $k_s$ at time $s$, for any $s < t$.

   (a) If $i = i'$ and $L = (i, j', k_s)$, then the two triangles overlap in the edge $i$, and $\widehat{S}'_{J\cap L, t\vee S} = 1/p_{i,t}$. Thus, the estimator of the covariance is,
   $$\mathrm{Cov}(\widehat{S}'_{J,t}, \widehat{S}'_{L,s}) = (p_{i,t}p_{j,t})^{-1}\big(p_{i,s}^{-1} - 1\big)p_{j',s}^{-1}$$
   Thus, for all triangles $L = (i, j', k_s)$, for $s < t$
   $$\sum_{s<t} \mathrm{Cov}(\widehat{S}'_{J,t}, \widehat{S}'_{L,s}) = (p_{i,t}p_{j,t})^{-1} \sum_{s<t} \big(p_{i,s}^{-1} - 1\big)p_{j',s}^{-1}$$
   $$= (p_{i,t}p_{j,t})^{-1} * U_{i,t}$$
   where $U_{i,t} = \sum_{s<t}\big(p_{i,s}^{-1} - 1\big)p_{j',s}^{-1}$

   (b) If $j = j'$ and $L = (i', j, k_s)$, then similar to the previous case, then the estimator of the covariance is,
   $$\mathrm{Cov}(\widehat{S}'_{J,t}, \widehat{S}'_{L,s}) = (p_{i,t}p_{j,t})^{-1}\big(p_{j,s}^{-1} - 1\big)p_{i',s}^{-1}$$
   Thus, for all triangles $L = (i', j, k_s)$, for any $s < t$
   $$\sum_{s<t} \mathrm{Cov}(\widehat{S}'_{J,t}, \widehat{S}'_{L,s}) = (p_{i,t}p_{j,t})^{-1} \sum_{s<t} \big(p_{j,s}^{-1} - 1\big)p_{i',s}^{-1}$$
   $$= (p_{i,t}p_{j,t})^{-1} * U_{j,t}$$

   where $U_{j,t} = \sum_{s<t}\big(p_{j,s}^{-1} - 1\big)p_{i',s}^{-1}$.

   (c) if $k_s = i$ or $k_s = j$, then the estimator of the covariance is zero,
   $$\mathrm{Cov}(\widehat{S}'_{J,t}, \widehat{S}'_{L,s}) = (p_{i,t}p_{j,t})^{-1}\big((p_{i',s}p_{j',s})^{-1} - (p_{i',s}p_{j',s})^{-1}\big) = 0$$

To facilitate incremental covariance computations for streaming data, we define $U_{i,t}$ and $U_{j,t}$ as the cumulative sum variables for edges $i$ and $j$ respectively, to keep track of previously sampled *triangle estimators* that contain $i$ and $j$ respectively, at any time $s < t$. Note that for the new arriving edge $k$, we have $U_{k,t} = 0$. Now, we add the covariance increments to each edge as follows,

$$\mathrm{Var}(\widehat{n}_i) \leftarrow \mathrm{Var}(\widehat{n}_i) + 2 * U_{i,t} * (p_{i,t}p_{j,t})^{-1}$$
$$\mathrm{Var}(\widehat{n}_j) \leftarrow \mathrm{Var}(\widehat{n}_j) + 2 * U_{j,t} * (p_{i,t}p_{j,t})^{-1}$$

Then, to update the cumulative variables for edges $i, j \in J = (i, j, k)$.
$$U_{i,t} \leftarrow U_{i,t-1} + \big(p_{i,t}^{-1} - 1\big)/p_{j,t}$$
$$U_{j,t} \leftarrow U_{j,t-1} + \big(p_{j,t}^{-1} - 1\big)/p_{i,t}$$

**Unbiased Estimator for $\mathrm{Cov}(\widehat{S}'_{J,t}, \widehat{I}'_{L,s})$.** By applying Theorem 4, we detail the computation of the covariance $\mathrm{Cov}(\widehat{S}'_{J,t}, \widehat{I}'_{L,s})$:

1. If $J \cap L = \emptyset$, then from Lemma 1, the $\mathrm{Cov}(\widehat{S}'_{J,t}, \widehat{I}'_{L,s}) = 0$.

2. If $s = t$ and $J = L$, then the $\mathrm{Cov}(\widehat{S}'_{J,t}, \widehat{I}'_{J,t}) = (p_{i,t}p_{j,t})^{-1} - 1$.

3. If $s = t$ and $J \neq L$, then $J \cap L = \{k\}$. And from Theorem 4 (I), the $\mathrm{Cov}(\widehat{S}'_{J,t}, \widehat{I}'_{L,t}) = 0$

4. If $s < t$, and $L = (i', j', k_s)$ is a triangle completed by edge $k_s$ at time $s$ then,

   (a) If $i = i'$ and $L = (i, j', k_s)$, then $J \cap L = \{i\}$, and the covariance estimator is,
   $$\mathrm{Cov}(\widehat{S}'_{J,t}, \widehat{I}'_{L,s}) = (p_{i,t}p_{j,t})^{-1}(1 - p_{i,s})$$

   And, for all triangles $L = (i, j', k_s)$, for any $s < t$,
   $$\sum_{s<t} \mathrm{Cov}(\widehat{S}'_{J,t}, \widehat{I}'_{L,s}) = (p_{i,t}p_{j,t})^{-1} \sum_{s<t}(1 - p_{i,s})$$
   $$= (p_{i,t}p_{j,t})^{-1} * D_{i,t}$$

   where $D_{i,t} = \sum_{s<t}(1 - p_{i,s})$.

   (b) if $j = j'$ and $L = (i', j, k_s)$, then $J \cap L = \{j\}$, the covariance estimator is, $\mathrm{Cov}(\widehat{S}'_{J,t}, \widehat{I}'_{L,s}) = (p_{i,t}p_{j,t})^{-1}(1 - p_{j,s})$.
   $$\mathrm{Cov}(\widehat{S}'_{J,t}, \widehat{I}'_{L,s}) = (p_{i,t}p_{j,t})^{-1}(1 - p_{j,s})$$

   And, for all triangles $L = (i', j, k_s)$, for any $s < t$,
   $$\sum_{s<t} \mathrm{Cov}(\widehat{S}'_{J,t}, \widehat{I}'_{L,s}) = (p_{i,t}p_{j,t})^{-1}(1 - p_{j,s})$$
   $$= (p_{i,t}p_{j,t})^{-1} * D_{j,t}$$

   where $D_{j,t} = \sum_{s<t}(1 - p_{j,s})$.

   (c) If $k_s = i$ or $k_s = j$, then the $\mathrm{Cov}(\widehat{S}'_{J,t}, \widehat{I}'_{L,s}) = 0$.

We define $D_{i,t}$ and $D_{j,t}$ as the cumulative sum variables for edges $i$ and $j$ respectively, to keep track of previously sampled *triangle indicators*, that contain $i$ and $j$ respectively, at any time $s < t$. Note that for the new arriving edge $k$, we have $D_{k,t} = 0$.

**Estimating the $\mathrm{Cov}(\widehat{S}'_{L,s}, \widehat{I}'_{J,t})$.** For $s < t$, the estimate of the $\mathrm{Cov}(\widehat{S}'_{L,s}, \widehat{I}'_{J,t})$ is similar to the cases discussed previously. Thus, we adopt the same form in Theorem 4 (I). Note that while Theorem 4 (I) does not treat this case, it is straightforward to show that the estimator is also unbiased for the $\mathrm{Cov}(\widehat{S}'_{L,s}, \widehat{I}'_{J,t})$. Hence, if $J \cap L = \{i\}$, the covariance estimator is,
$$\mathrm{Cov}(\widehat{S}'_{L,s}, \widehat{I}'_{J,t}) = (p_{i,s}^{-1} - 1)p_{j',s}^{-1}$$

Thus, for all triangles $L = (i, j', k_s)$ and $s < t$,
$$\sum_{s<t} \mathrm{Cov}(\widehat{S}'_{L,s}, \widehat{I}'_{J,t}) = \sum_{s<t}(p_{i,s}^{-1} - 1)p_{j',s}^{-1} = U_{i,t}$$

Similarly, if $J \cap L = \{j\}$, the covariance estimator is,
$$\mathrm{Cov}(\widehat{S}'_{L,s}, \widehat{I}'_{J,t}) = (p_{j,s}^{-1} - 1)p_{i',s}^{-1}$$

And, for all triangles $L = (i', j, k_s)$ and $s < t$,
$$\sum_{s<t} \mathrm{Cov}(\widehat{S}'_{L,s}, \widehat{I}'_{J,t}) = \sum_{s<t}(p_{j,s}^{-1} - 1)p_{i',s}^{-1} = U_{j,t}$$

Now, we add all the covariance increments for each edge as follows,

$$\mathrm{Cov}(\widehat{n}_i, w_i) \leftarrow \mathrm{Cov}(\widehat{n}_i, w_i) + \left(p_{i,t}p_{j,t}\right)^{-1} - 1 + D_{i,t} * \left(p_{i,t}p_{j,t}\right)^{-1} + U_{i,t}$$

$$\mathrm{Cov}(\widehat{n}_j, w_j) \leftarrow \mathrm{Cov}(\widehat{n}_j, w_j) + \left(p_{i,t}p_{j,t}\right)^{-1} - 1 + D_{j,t} * \left(p_{i,t}p_{j,t}\right)^{-1} + U_{j,t}$$

$$\mathrm{Cov}(\widehat{n}_k, w_k) \leftarrow \mathrm{Cov}(\widehat{n}_k, w_k) + \left(p_{i,t}p_{j,t}\right)^{-1} - 1$$

Then, to update the cumulative variables for edges $i, j \in J = (i, j, k)$.

$$D_{i,t} \leftarrow D_{i,t-1} + \left(1 - p_{i,t}\right)$$

$$D_{j,t} \leftarrow D_{j,t-1} + \left(1 - p_{j,t}\right)$$

We summarize all the variance and covariance computations in Algorithm 2, which is a supplementary to Algorithm 1 (in the case of triangle motifs).

---

**Algorithm 2** Iterative Variance Computation Following Line 14 in Algorithm 1

---

**Input:** New edge $k$, current sample set $\widehat{K} \ni k$, triangle $h = (j_1, j_2, k) \subset \widehat{K}$, $p(h) = p(j_1)p(j_2)$

  **for** edge $j \in h$ **do**
    $\mathrm{Var}(j) \leftarrow \mathrm{Var}(j) + \left(p(h)^{-1} - 1\right)/p(h)$
    $\mathrm{Cov}(j) \leftarrow \mathrm{Cov}(j) + p(h)^{-1} - 1$
  **for** $j \in h : j \neq k$ **do**
    $\mathrm{Var}(j) \leftarrow \mathrm{Var}(j) + 2 * U(j)/p(h)$
    $\mathrm{Cov}(j) \leftarrow \mathrm{Cov}(j) + U(j) + D(j)/p(h)$
    $U(j) \leftarrow U(j) + \left(p(j)^{-1} - 1\right)/p(j'), \{j'\} = h \setminus \{j, k\}$
    $D(j) \leftarrow D(j) + 1 - p(j)$

---

## C   Ablation Study

To understand the effects of the various design choices in the proposed framework APS with shrinkage estimation, we conduct a thorough set of ablation study experiments. The proposed APS method provides a sampling framework that consists of two major parts: (1) Adaptive sampling with importance weights, and (2) James-Stein shrinkage estimation. Hence, there are several design choices to make, e.g., we could choose to use adaptive sampling *without* shrinkage estimation.

Results in Table 2 clearly show that shrinkage estimation significantly improves the performance of APS sampling. Another design choice is to use non-adaptive priority sampling where the edge weights/ranks are computed once at the time of sampling, and fixed during the rest of the streaming process. We conducted this experiment on the same datasets by using only the sampling weights assigned at arrival time (Line 12 in Alg 1) and fix it for the rest

Table 4: MSE for Non-Adaptive Sampling ($f = 0.2$)

| graph | Non-Adapt | Non-Adapt (JS) |
|---|---|---|
| SOC-FLICKR | 4907.21 | **2174.9** |
| SOC-LIVEJOURNAL | 94.46 | **69.97** |
| SOC-YOUTUBE-SNAP | **24.78** | 31.704 |
| WIKI-TALK | **78.69** | 98.765 |
| WEB-BERKSTAN-DIR | 1723.63 | **1236.3** |
| CIT-PATENTS | 6.45 | **5.67** |
| SOC-ORKUT-DIR | 405.86 | **227.65** |

of the stream. We summarize the results in Table 4. For some graphs (e.g., soc-flickr), we observed that using non-adaptive weights in APS might perform better than using adaptive weights.

We conjecture this is due to the excessive variance of APS in the estimated count of the edges with small triangle counts, and can be observed in the tail of the distribution (see Figure 7). However, among all the design choices, the combination of (APS sampling + adaptive weights + shrinkage estimation) has the strongest regularization effect on the performance of graph sampling. We also observe that applying the shrinkage estimator to the non-adaptive sampling significantly improve the performance. These effects are demonstrated in Figures 6 and 7 which show the distribution of non-adaptive APS and adaptive APS respectively (with and without shrinkage estimation).

In summary, the results suggest that APS with shrinkage performs significantly better than related methods in previous work, and each of the design choices contributes to the final performance.

Figure 6: Sample size $f = 0.4$. Left: Non-adaptive Priority Sampling, estimate vs exact. Right: Non-adaptive Priority Sampling with Shrinkage estimator (James-Stein JS) vs exact.

Figure 7: Sample size $f = 0.4$. Left: Adaptive Priority Sampling, estimate vs exact. Right: Adaptive Priority Sampling with Shrinkage estimator (James-Stein JS) vs exact.

# D    Additional Plots

Figure 8: Each Plot corresponds to one graph at sampling fractions $f = \{0.20, 0.40\}$, and shows the raw count of the top-1M edges using Uniform Sampling [53] vs the actual count. The top-1M edges are ranked based on their true counts. $x$-axis: the rank of top edges 1–1M in $\log_{10}$ scale, $y$-axis: weights.

Figure 9: Each Plot corresponds to one graph at sampling fractions $f = \{0.20, 0.40\}$, and shows the raw count of the top-1M edges using Triest sampling [48] vs the actual count. The top-1M edges are ranked based on their true counts. $x$-axis: the rank of top edges 1–1M in $\log_{10}$ scale, $y$-axis: weights (triangle count per edge).

Figure 10: Each Plot corresponds to one graph at sampling fractions $f = \{0.20, 0.40\}$, and shows the normalized count of the top-10K edges using APS with Shrinkage Estimation vs the actual normalized count. The top-10K edges are ranked based on their true normalized counts. The $x$-axis: the rank of top edges 1–10K in $\log_{10}$ scale, the $y$-axis: normalized weights.

# E  Dataset Details

- **soc-flickr**: Crawl of the Flickr photo-sharing social network from May 2006. Nodes are users and edges represent that a user added another user to their list of contacts [19].
- **soc-livejournal**: LiveJournal is an online social community publishing platform, Nodes are users and edges are user-to-user links [35].
- **soc-youtube**: Youtube social network. Nodes are users and edges are user-to-user friendship links [35].
- **wiki-Talk**: Wikipedia network of user discussions from the inception of Wikipedia till January 2008. Nodes are Wikipedia users and edges are user-to-user edits of talk pages [31].
- **web-BerkStan-dir**: Web network where nodes represent webpages from Berkely and Stanford and edges represent hyperlinks among them [30].
- **cit-Patents**: The citation graph of US Patents includes all citations made by patents granted between 1975 and 1999 [29].
- **soc-orkut-dir**: Orkut online social network, where nodes represent users and edges represent user-to-user friendship links [35].