[Reviews · NeurIPS 2020]

Review 1

Summary and Contributions: The paper proposes a heuristic method for motif (subgraph) counting in graphs in a streaming setting. Most commonly, triangles or small cliques. In the streaming setting, edges arrive in a stream and a fraction of the edges is samples and retained. The number of subgraphs can then be estimated from the number in the sample set, and for unbiasedness, adjusted by the product of the sampling probabilities of the participating edges. The method proposed in this paper is based on the assumption that edges that at the point processed participate in more subgraphs are more likely to participate in more subgraphs as more edges are added. Therefore, they should be retained with higher probability than other edges. Unbiasedness can still be ensured as long as the "inclusion probability" of each edge is properly tracked.

Strengths: + Important and well studied problem + Very nice presentation. + The proposed heuristic is principled and natural. + Experiments demonstrate that this sampling heuristic is highly effective on large graphs. Helpful and extensive validity analysis of the approach.

Weaknesses: I did not identify significant weaknesses. I enjoyed the submission.

Correctness: As far as I could tell. I did not verify each detail.

Clarity: Yes.

Relation to Prior Work: As far as I could tell.

Reproducibility: Yes

Additional Feedback:


Review 2

Summary and Contributions: The authors propose an approach to estimate the motif adjacency matrix of a graph in a streaming data setting. Each entry (i,j) in the motif adjacency matrix measures the number of co-occurrences of the two nodes i and j in a specified motif, e.g. a triangle. The authors propose an unbiased estimator of the motif adjacency matrix entries in the streaming setting using adaptive priority sampling (APS). They then propose a shrinkage estimator in the style of James-Stein to reduce the mean-squared error (MSE) of estimation by reducing variance at the cost of introducing some bias. The proposed estimators are shown to significantly outperform baseline methods both in MSE and relative spectral norm on 7 large network data sets with millions of nodes.

Strengths: + Using James-Stein shrinkage in this problem setting is innovative and novel. + Significant performance improvement in both APS and APS + shrinkage compared to baselines.

Weaknesses: - Presentation can be significantly improved. See my comments below on clarity. - Breadth of experiments is low. I was expecting to see some additional experiments in the supplementary material on motifs other than triangles. Also, the experiments use only static graphs in a simulated streaming setting by randomly choosing the order of edge arrivals. Even one demonstration of its performance on a small timestamped graph, e.g. the Facebook WOSN 2009 data (with ~90k nodes and ~3.6M edges), would create a more realistic experiment. Viswanath, B., Mislove, A., Cha, M., & Gummadi, K. P. (2009). On the evolution of user interaction in Facebook. In Proceedings of the 2nd ACM Workshop on Online Social Networks (pp. 37-42): ACM Press. Edit after author response: The authors demonstrated their APS both with and without shrinkage on the Facebook WOSN 2009 data with strong results, so I am less concerned about this weakness.

Correctness: I did not identify any technical issues, although I did not read the theoretical analysis in great detail.

Clarity: Plots are way too small to be readable. This is part of a greater problem--there is too much content that is compressed into the 8 page limit. For example, the derivation of the optimal shrinkage coefficient lambda can probably be moved to the supplementary. Figure 1 could probably be moved also, and Section 2.3 could be moved while simply stating the computational benefit.

Relation to Prior Work: I am not very familiar with the prior work on graph sampling for higher-order network structures, but the references seem to be plentiful. Some more references to uses of James-Stein shrinkage in other settings like covariance or affinity matrix estimation (Ledoit & Wolf, 2003; Schäfer & Strimmer, 2005; Chen et al., 2010; Xu et al., 2014) in addition to ref. [9] in the paper could be useful to readers as it may not be a well-known concept to much of the NeurIPS audience. Ledoit, O., & Wolf, M. (2003). Improved estimation of the covariance matrix of stock returns with an application to portfolio selection. Journal of Empirical Finance, 10(5), 603-621. Schäfer, J., & Strimmer, K. (2005). A shrinkage approach to large-scale covariance matrix estimation and implications for functional genomics. Statistical Applications in Genetics and Molecular Biology, 4(1). Chen, Y., Wiesel, A., Eldar, Y. C., & Hero, A. O. (2010). Shrinkage algorithms for MMSE covariance estimation. IEEE Transactions on Signal Processing, 58(10), 5016-5029. Xu, K. S., Kliger, M., & Hero III, A. O. (2014). Adaptive evolutionary clustering. Data Mining and Knowledge Discovery, 28(2), 304-336.

Reproducibility: Yes

Additional Feedback: Line 105: (2) how to an unbiased estimate: missing a verb between to and an Line 166: P_{J,t} it computed: it -> is Figures being referred to in the supplementary appear to be off by 1, e.g. the main body text refers to Figure 6 when it should be Figure 7 and Figure 9 when it should be Figure 10. The authors' statement of broader impact is about impact to research, not broader impact to society. Edit after author response: I increased my score slightly in response to the additional results shown in the author response. There are still several weaknesses in the paper, primarily the clarity and presentation, but I find it to be a good contribution overall.


Review 3

Summary and Contributions: The authors propose an algorithm that estimates the number of motifs (say triangles) adjacent to each edge. The algorithm is based on maintaining a sample of edges and using the number of triangles incident to each edge within the sample for an estimator for the true number. The sample is a weighted version of priority sampling.

Strengths: The authors ran their algorithm on a large number of data sets. The main idea may have merit.

Weaknesses: The writing is not clear at all. The math is not stated clearly, and the complexity analysis is not satisfactory. I suspect the algorithm is actually quite memory intensive, to a point where it defeats the purpose of sampling. The supp material does not contain code to check for myself.

Correctness: It is hard for me to verify correctness.

Clarity: The introduction is reasonably well written but the technical parts are not clear.

Relation to Prior Work: The authors mentions relevant related work, but they don't explain exactly how they differ. For instance, algorithm Triest which is relevant is only mentioned in the experimental section.

Reproducibility: No

Additional Feedback: The most important remarks is that there is no good analysis of the complexity of the algorithm, and that some aspects of what the algorithm guarantees are not clear. See below. My comments are by the line number in the paper, some are substantial, some are mere style or minor corrections. 30: From the introduction one can not understand what 'high order properties' are or how they differ from other graph properties. The algorithm aims to count subgraphs, or as it is called in some communities 'motifs'. 46: I suggest around here you actually state what exactly is the problem you aim to solve. 48-64: I think you conflate streaming algorithms with graph sparsification. Graph sparsification aims at finding a sparse sub graph that could be used to estimate properties of the original (possibly dense) graph, and is a large field of study. It often uses fairly sophisticated methods of sampling to do so. Streaming algorithms typically use *constant* or at most *log* size memory as it streams through the input to compute something. Your algorithm streams through the input, and create a sparse graph. Your memory footprint is actually quite big. 71: I don't understand figure 1. What are the units on the axis? what do the lines represent? 99: probably not the right place to insert a pointer to the supp material. 100: It's a dangling sentence, is it a typo? 105: missing word? 126: This could be expensive time-wise no? Each motif might be scanned multiple times. There is no proper analysis of the running time. 136: When k is removed then all it's associated variables are thrown out too. Correct? 148: Just say that S_i,t is the indicator of the event that edge i arrived before time t. 168: I can't understand what Theorem 1 says. 174: The estimator is unbiased only for edges in the sample! There are n^2 possible edges. Edges outside the samples are of course 0. You need some sort of analysis about the probability an edge stays in the sample, or some such thing. This most likely depends on the order of arrival of the edges. So it is not clear at the end of the day what is the guarantee of the algorithm. 192: Aren't the weights always non-decreasing? I'm confused. 202: The analysis of complexity is lacking. The running time depends on the number of triangles in the graph clearly, but each triangle might be computed multiple times. 360: I'm not sure I understand the comparison with Triest. Triest estimates the number of triangles per node no? Update: In view of author response I raised my score. I still think the technical presentation is lacking. The main theorems are very hard to follow, the complexity analysis is not precise etc. I encourage the authors to improve the paper before the final submission (if it is to be accepted).


Review 4

Summary and Contributions: This paper presents an adaptive variant of reservoir sampling for streams of edges to estimate the edge-local counts of small subgraphs. It also proposes a shrinkage estimator that allows to trade-off bias for reduced variance, thus improving the repeated-sampling performance of the algorithm.

Strengths: * Nice mathematical work that tackles the essence of the problem * Interesting experimental results * Important problem studied

Weaknesses: * Unclear comparison with some baseline

Correctness: The theoretical results seem correct, and the experimental evaluation feels convincing.

Clarity: The paper is a little heavy on the mathematical side, but that's actually a plus. It could perhaps benefit from some more intuition (there are examples and so on in the supplementary materials, but that would require the reader to interrupt the reading and look into the supplementary material)

Relation to Prior Work: Yes, the previous work is described appropriately.

Reproducibility: No

Additional Feedback: This paper is interesting and well written and it tackles an important problem in a novel and elegant way. The use of weighted adaptive sampling is ingenious and something that was missing in previous work, in part because they were focused only partially on the local counts. There is a complete mathematical analysis of the approach and a nice experimental evaluation. In some parts, the paper feels a bit too dense. Given the large use of supplementary materials, perhaps some reorganization can be made to give more intuition in the main text. How is \phi (line 4 of algo 1) chosen, and what is the impact of different choices on the performances of the algorithm? The comparison with the Triest paper is nice, but it is unclear which version of Triest was used. Line 166: Typo: "it computed" -> "is computed" Please do not use italic for "e.g." and "i.e.", as per most manuals of style. ## AFTER AUTHORS' RESPONSE AND DISCUSSION WITH OTHER REVIEWERS I thank the Authors for replying to my comments in an acceptable way.

[Author Response · NeurIPS 2020]

# Review Response: "Adaptive Shrinkage Estimation for Streaming Graphs"

**We thank all the reviewers for their time and feedback. The main reservations of the reviewers are addressed below, and all requested clarifications will be added to the final version.**

## Citations & references (Reviewer #2)

Thank you for pointing us to this work, we completely agree, and have now included references to the works by Ledoit & Wolf, 2003; Schäfer & Strimmer, 2005; Chen et al., 2010; Xu et al., 2014.

## Demonstration on a timestamped graph (Facbeook WOSN 2009) (Reviewer #2)

Thanks for pointing us to this. As suggested, we provide a demonstration on a timestamped graph.

**Spectral error:** APS JS (with Shrinkage): $0.04$, APS: $0.31$, Uniform Sampling: $0.40$, Triest: $0.58$.

## How is $\phi$ (line 4 of algo 1) chosen, and what is the impact of different choices (Reviewer #4)

We use $\phi$ in Alg 1 to denote the initial weight of an edge, such that $\phi > 0$. This initial weight is required to guarantee that each edge is assigned a non-zero probability. We chose $\phi = 1$ to be comparable with the edge weight increments due to subgraphs incident to each edge. This procedure allows edges to have a chance to be included in the sample with a non-zero probability, regardless of the number of subgraphs incident to them, but not so large as to damp out their topological weight.

## Comparison with Triest is nice, but it is unclear which version of Triest used. (Reviewer #4)

We compare with Triest-IMPR [40], the improved Triest algorithm with higher estimation quality.

## Highlighting Algorithm & Complexity Analysis (Reviewer #3)

- Reviewer #3: "The running time depends on the number of triangles in the graph clearly, but each triangle might be computed multiple times."

There may be some confusion here. The algorithm uses a *small fixed-size memory* to store the sample $(m = |\widehat{K}|)$. The first $m$ edges are added to the sample. Then, each subsequent edge is *provisionally* included in the current sample from which an edge is discarded along with its associated variables (i.e., to maintain the sample size $m$). When a new edge arrives in the stream, it is processed *only once*, and each motif completed by the new edge whose other edges are in the current sample is also processed only once. See Sec 2.3 (line 202) for the cost analysis of all sample updates (insert/delete/update).

## Emphasizing Algorithm Guarantees (Reviewer #3)

The algorithm is guaranteed to provide the unbiasedness property for the reasons stated below.

- Reviewer #3: "The estimator is unbiased only for edges in the sample."

There is some confusion. Edges are included in the sample based on their assigned probabilities. Thus, edges not retained in the sample indeed have zero estimator. Edges that are included in the sample have an estimate which is reciprocal of their probability to be retained in the sample. Unbiasedness is a general property of inverse probability estimators that are used in numerous areas of computer science & statistics, including graph sampling; e.g. reference [1] in the paper. This procedure guarantees unbiased estimators for the entire graph (i.e., $\mathbb{E}[\widehat{A}] = A$, A denotes adjacency matrix).

- Reviewer #3: "You need some sort of analysis about the probability an edge stays in the sample."

Yes, this is precisely what we do (see Sec 2.2): for each edge in the sample, we compute the probability that it has been retained so far (Alg 1, line 11); the inverse probability estimate for that edge is the reciprocal of that probability. Theorem 1 says that the probability of a subgraph is the product of the probabilities of its edges (computed in Alg 1, line 13), and finally, the corresponding unbiased estimator for that subgraph is the reciprocal of that product, which gets added into the estimated count of subgraphs incident to edges (Alg 1, line 14). Theorem 1 states the inverse probability estimator for subgraph counts and proves it is unbiased (see lines 165-167 for details). It shows how the estimators can be computed in practice from random/state variables during stream processing.

- Reviewer #3: "Aren't the weights always non-decreasing?"

Theorem 1 applies for any weights, non-decreasing or not. In our case, the weights are non-decreasing, and Theorem 2 shows the computational benefit results from this property, which we exploit in Alg 1.

## Typos/Presentation (Reviewer #2 & Reviewer #4)

Last, we thank all reviewers for pointing out a few typos and have now fixed them. Reviewer #2 & #4 suggested to move a few parts to supplementary materials. As suggested, we will move the derivation of the optimal shrinkage coefficient $\lambda$ in section 3.1, and Section 2.3 to the supplementary materials.

[Meta-Review · NeurIPS 2020]

The paper proposes a streaming algorithm for finding graph motifs, using priority sampling and shrinkage estimators. Four knowledgeable reviewers carefully considered the paper, and they all recommend to accept. Therefore, I also recommend to accept the paper for NeurIPS 2020. However, I strongly encourage the authors to address all the issues raised by the reviewers in the final version, and additionally to provide software for reproducibility of the empirical results.